# Serological and Uterine Biomarkers for Detecting Endometritis in Mares

**DOI:** 10.3390/ani13020253

**Published:** 2023-01-11

**Authors:** Stefano Cecchini Gualandi, Tommaso Di Palma, Raffaele Boni

**Affiliations:** 1Department of Sciences, University of Basilicata, Campus Macchia Romana, 85100 Potenza, Italy; 2Veterinarian Private Practitioner, 85056 Ruoti, Italy

**Keywords:** mare, endometritis, serological biomarkers, uterine fluid biomarkers, oxidative stress, ferric reducing ability of plasma (FRAP)

## Abstract

**Simple Summary:**

In the mare, infectious/inflammatory pre-breeding endometritis may be exacerbated by post-breeding endometritis (PBE), which occurs due to the reactivity of the uterus to insemination. PBE can be treated post-insemination; however, it is difficult to monitor the outcome of the treatment due to negative interference in embryonic development following the collection of uterine samplings. Indirect findings capable of providing information on the evolution of this pathology and verifying the response to treatments, therefore, would be highly recommended. This study examined ten parameters in blood and uterine fluid associated with the (anti)oxidant status, inflammation, and protease regulator potential and possible markers of endometritis in the mare. Endometritis has been evaluated with both culture and cytological techniques; the latter has been carried out by both classical and fluorescence techniques. Among the parameters examined in the blood serum, only one of them, the ferric reducing ability of plasma (FRAP) assay, a marker of the antioxidant power, allowed significant discrimination between the mares diagnosed as cytologically positive and negative for endometritis. This result provides an important piece of information in the development of indirect methodologies for the diagnosis of endometritis in the mare.

**Abstract:**

Serological analysis may provide relevant information on endometritis diagnostics. Therefore, mares scheduled for AI with refrigerated semen, at the time of heat signs, underwent blood and uterine fluid samplings using a swab, uterine lavage for culture analysis, and treatment with human chorionic gonadotropin to induce ovulation. After 24–28 h, the mares were inseminated and, if positive at the culture test, treated with antibiotics chosen based on the susceptibility test. Uterine cells obtained by swabs were used for cytological examination with both classical and fluorescence techniques. Blood serum and uterine fluid samples were analyzed for assessing parameters related to redox balance, inflammation, and protease regulator potential. In blood serum, total antioxidant capacity, measured as the ferric reducing ability of plasma (FRAP), was significantly lower in cytologically endometritis-positive than -negative mares. In the uterine fluid, total thiol levels (TTL), nitric oxide metabolites (NO_x_), protease activity and total protein content varied significantly between groups. Although the cytological examination was more capable of discriminating between endometritis-positive and -negative mares in relation to the parameters examined, no statistically significant differences emerged in terms of pregnancy rate in relation to cytological and culture diagnosis as well as in mares diagnosed as positive and negative for endometritis.

## 1. Introduction

An accurate diagnosis of endometritis is an essential requirement for an efficient reproductive activity and the application of reproductive technologies. Subclinical infectious endometritis has been found to affect 48.8% of clinically healthy mares before breeding [1]. Such a situation may be further aggravated by post-breeding endometritis (PBE) resulting from the inflammation provoked by introducing fresh or frozen semen at insemination [2,3]. Normally, the endometrium solves this inflammation in few hours; however, if the uterine fluid is present at 12 h or more after insemination, there is a persistent breeding-induced endometritis [4]. The possibility for the uterine mucosa to overcome this noxa is related to individual susceptibility associated with numerous factors such as age, the conformation of the perineum, the position of the uterus in the pelvic cavity, and the excess fluid retention during diestrus [5,6]. Often, the anamnesis indicating animals with a history of low fertility is helpful [7]. PBE can be easily detected when it is characterized by uterine fluid retention by means of clinical and ultrasound examination [8]. However, the presence of fluid in the uterus indicates acute inflammation rather than bacterial infection. Hence, uterine fluid retention does not represent a reliable sign of infectious endometritis, reducing the possibility that this pathology would be detected with clinical practices alone [9]. Laboratory techniques based on microbiological (or culture) and cytological tests of samples taken from the uterus were, thus, developed to improve the diagnostic accuracy. The culture of uterine samples for microbiological assessment has good diagnostic reliability but requires a correct executive procedure to ensure the hygiene of the collection in all its phases, and one or two days are needed for the growth in culture of the microorganisms taken from the uterine site [10]. The cytological examination allows faster responses if based on a smear of uterine cells collected by cytobrush, cytotape, swab, or low-volume uterine washing techniques [11,12,13]. However, cytological preparations are not always easy to read and require a high level of experience. To facilitate the application of this technique, we have recently introduced the use of the fluorescence spectrometry technique [14], based on the possibility of marking polymorphonuclear leukocytes (PMNs), i.e., the main cells of the immune system involved in the pathogenesis of endometritis, with fluorescent dyes. PBE can be treated post-insemination; however, it is difficult to monitor the outcome of the treatment, considering that sampling the uterine material on >3 days post-breeding may have repercussions for subsequent embryonic development. The possibility of using indirect markers, which do not require uterine samples, for the diagnosis of endometritis in the mare was recently verified by Abdelnabi and colleagues [15] who evaluated some oxidative stress (OS) parameters in blood serum such as malondialdehyde (MDA), total antioxidant capacity (TAC), and nitric oxide metabolites (NOMs). While NO_x_ and MDA were significantly (*p* < 0.05) higher in cytologically endometritis-positive than -negative mares, TAC values showed an inverse trend. Another study performed on endometritis-susceptible mares at foal heat [16] found that neutrophil activity increased together with MDA and fibrinogen plasma levels, whereas myeloperoxidase (MPO) activity was slightly lower in endometritis-susceptible than in -resistant mares. In thoroughbred mares receiving a dietary supplementation with antioxidants, such as α-tocopherol, during the peripartum period, FRAP levels in the blood serum did not differ between treated and control groups but were higher in mares >10 years old [17]. However, serum reactive oxygen metabolites (ROMs) did not differ in relation to the number of services to get pregnant.

Along these lines, it seems interesting to develop alternative diagnostic approaches to detect uterine inflammation without a direct intervention on the uterus. This topic was investigated in a recent review aimed at gathering information on the equine and bovine species of blood biomarkers linked to oxidative stress and associated with endometritis occurrence [18]. In the present study, the blood levels of markers associated with redox balance, inflammation, and protease regulator potential have been assessed in mares at the time of estrus and before insemination in order to highlight possible diagnostic evidence of endometritis without accessing the endometrium. These findings were compared with those obtained from similar uterine fluid sample analyses. The presence of endometritis was evaluated by using both bacteriological and cytological evaluations. The latter was conducted on cells collected by uterine swab by either microscopy evaluation of cytology smears or fluorescence microscopy.

## 2. Material and Methods

### 2.1. Reagents

If not otherwise indicated, all reagents and media were purchased from Sigma-Aldrich (Milan, Italy) and all cell culture tested.

### 2.2. Animals

The study involved 33 quarter horse mares scheduled for artificial insemination (AI) with refrigerated semen that were enrolled on a first-come, first-served basis at a stud farm (Service 3D, Avigliano, Italy). The mares had a mean (±SD) age of 10.3 ± 4.9 years, 1.6 ± 2.3 number of births, and did not show any particular health or reproductive problems detected during clinical analysis that would require special attention. The mares were housed in box stalls in natural light conditions and fed mixed meadow hay ad libitum, 1 kg/day oat seeds, and 1 kg/day integrated compound feed for horses (Equimix, Specialmix Miglionico s.r.l., Altamura, Italy); they were client-owned and informed owner consent was obtained.

At the time of behavioral signs of estrus, together with the presence of a large ovarian follicle ≥ 35 mm in diameter and uterine edema, the mares underwent blood and uterine fluid sampling using a swab, uterine lavage for culture analysis, and treatment with 2000 IU human Chorionic Gonadotropin (Chorulon, MSD Animal Health, Milan, Italy) to induce ovulation. After 24–28 h, the mares were inseminated and, if the culture test was positive, treated 24–72 h post-insemination with antibiotics based on the result of the susceptibility test. At this time, before antibiotic treatment and, in general, in all inseminated mares, a uterine flushing with 1 L of Ringer’s lactate was carried out to alleviate the post-insemination inflammatory response and eliminate inflammatory elements and dead spermatozoa [3,19]. Pregnancy detection was performed ultrasonographically between days 17 and 18 after insemination.

### 2.3. Ethics Approval

This study has been conducted according to the guidelines of the European Directive 63/2010 on the protection of animals used for scientific purposes, transposed into the Italian law by Legislative Decree 2014/26. Considering that the proposed experimental design does not fall within the European Directive 63/2010, the Ethics Committee of the University of Basilicata (OpBA) has established that it did not require any authorization for it being performed.

### 2.4. Collection of Blood and Uterine Fluid Samples

At the time of ovulatory synchronization, the mares in heat were restrained in stocks. Blood samples were collected from the jugular vein with an 18-gauge needle in 10 mL sterile Vacutainer^®^ (Becton Dickinson Italia S.p.A., Milan, Italy) tubes without anticoagulants and stored in a thermal box at 4 °C. After that, their perineal regions were carefully cleaned and disinfected with povidone–iodine scrub. Then, a double-guarded uterine swab (IMV Biotechnologies Italia, Piacenza, Italy) was introduced through the vagina and cervix into the uterus. The swab was rotated in the uterine corpus and kept in contact with the endometrial surface for at least 30 s. Immediately after extraction from the vagina, the swab tip was inserted into a conical centrifuge tube containing 2 mL of sterile PBS (Life Technologies, Milan, Italy), and the swab shaft was cut allowing the closure of the tube. The samples thus obtained were immediately stored in a thermal box at 4 °C. Next, a uterine catheter was manually introduced through the vagina into the uterus that was infused with 500 mL of sterile Ringer’s lactate (Laboratorio Farmacologico S.A.L.F., Bergamo, Italy). The Ringer’s lactate flask was positioned at the bottom to recover the infused liquid by gravity flow. The collected samples were stored at 4 °C and transported to the laboratories of the Department of Sciences, University of Basilicata. Serum samples were separated after centrifugation at 2000× *g* for 20 min at 4 °C and stored in 1 mL aliquots at −80 °C until analysis. The flasks containing the uterine washing fluid were used for the culture test. The tubes containing the individual swabs were vortexed for 1 min, after which the swab was removed from the tube, which was centrifuged for 10 min at 500× *g* and 4 °C. The supernatant liquid was aliquoted into 1 mL eppendorf tubes and promptly frozen at −80 °C. The cell pellet was resuspended in 20 µL of PBS. A part of it was smeared on a slide to carry out the traditional cytological examination [20]. The remaining part was incubated with fluorescent dyes for fluorescence cytological assessment, as previously reported [14] with little modifications.

### 2.5. Microbiological Analyses

Uterine flushing samples were tested by conventional bacteriological detection, as previously reported [10,14,21]. In brief, the samples were inoculated in enrichment broth (Brain–Heart infusion) and incubated at 37 °C [21]. After 12–24 h, sowing was carried out on MacConkey and BD Columbia CNA agar plates enriched with 5% sheep blood and incubated at 37 °C for the isolation of Gram negative and positive bacteria, respectively. The inoculated plates were inspected for bacterial growth after 24 and 48 h. Single colonies of positive plates were sown on Mueller–Hinton agar plates (Liofilchem^®^, Roseto degli Abruzzi, Italy) and antibiotic discs were applied directly to the agar surface for the susceptibility test, following the Clinical and Laboratory Standard Institute procedures [22]. Plates were checked at either 12 or 24 h and the antibiotic showing the greatest inhibition halo diameter was considered the antibiotic of choice for the treatment of that uterine infection.

### 2.6. Cytological Analysis

Uterine cells obtained by swabs were used for cytological examination with both the classical and fluorescence techniques. Cell smears on slides were obtained as previously reported [14]. In brief, a concentrated cell suspension was dispersed on a slide, air-dried, fixed, and stained with May–Grünwald stain solution (0.25% (*w*/*v*) in methanol). Next, distilled water was deposited on the slide and, 3 min later, removed and replaced with a 10% diluted Gimsa solution. After 15 min, the slide was washed with tap water, air-dried, and observed at the microscope. The finding of one or more PMNs in a hundred counted cells labeled the cell sample as belonging to an endometritis-positive mare.

### 2.7. Fluorescent Cytological Analysis

The presence of PMN within the cell samples was detected by 2′,7′-dichlorodihydrofluorescein diacetate (H_2_DCF-DA) (Life Technologies, Milan, Italy) [14], whereas bis-benzimide trihydrochloride (H33342) (Life Technologies, Milan, Italy) was used to label cell nuclei to normalize PNM number over the total cell number of the sample. A 10 µL aliquot of each cell suspension obtained from uterine smears was incubated with 88 µL of PBS supplemented with 0.1% (*w*/*v*) PVA (PBS-PVA) and treated with 1 µL H_2_DCF-DA and 1 µL H33342 stock solutions. The former was prepared by diluting H_2_DCF-DA in DMSO (10 mM) and further diluted 1:10 in PBS at working time; the latter by diluting H33342 in distilled water (0.5 mM). Both stock solutions were stored in aliquots at −80 °C. After 30 min incubation, samples were centrifuged at 200× *g* for 5 min and washed twice in PBS-PVA using the same centrifugation protocol, as above. Then, cell pellets were resuspended in 10 µL of PBS-PVA and 3 µL of this cell suspension was laid on a microscope glass within a paraffin wax circle and slightly flattened with a coverslip. The H_2_DCF was oxidized by H_2_O_2_ to the highly fluorescent 2′,7′-dichlorofluorescein (DCF) and entrapped in the PMN. It was excited at 490 nm wavelength and emitted fluorescence at ~520 nm wavelength. The H33342 is a permeant fluorescent nuclear dye that was excited at 360 nm and emitted fluorescence at 450 nm wavelength. Samples were examined with a fluorescence microscope (EVOS FLoid Cell Imaging Station, Life Technologies, Carlsbad, CA, USA). The finding of one or more DCF-fluorescent cells on a hundred counted nuclei labeled the cell sample as belonging to an endometritis-positive mare.

### 2.8. Biochemical Analyses

The ferric reducing ability of plasma (FRAP) assay, based on the reduction of ferric ion to ferrous ion of the iron-Tris(2-pyridyl)-s-triazine complex, was determined as described by Benzie and Strain [23]. The assay was calibrated with iron (II) sulphate heptahydrate (FeSO_4_•7H_2_O) and the results are expressed in terms of FeSO_4_•7H_2_O equivalents (μM).

Total antioxidant capacity (TAC), based on the reduction of colored 2,2′-azinobis-(3-ethylbenzothiazoline-6-sulfonic acid) radical cation (ABTS^•+^), was measured according to Erel [24]. The assay was calibrated with ascorbic acid (AA) and the results are presented as AA equivalents (μM).

Free Radical Scavenging Activity (FRSA) was analyzed using the 2,2-Di(4-tert-octylphenyl)-1-picrylhydrazyl (DPPH) reduction assay, based on the reduction of DPPH· to 1,1-diphenyl-2-picryl hydrazine, as described by Blois [25], with minor modifications [26]. The assay was calibrated with AA and the results are presented as AA equivalents (mg mL^−1^).

Total thiol levels (TTL), based on the interaction of sulfhydryl groups (-SH) with 5,5-dithiobis-(2-nitrobenzoic acid) (DTNB) to form a colored anion, were measured as indicated by Hu [27]; the obtained data are presented in terms of TTL concentration (µM).

Total oxidant status (TOS) was analyzed using the original method based on the oxidation of ferrous ion to ferric ion and assessed by xylenol orange [28]. This assay was calibrated with tert-butyl hydroperoxide (*t*-BHP) and the results are presented as *t*-BHP equivalents (μM).

Nitric oxide radical (NO) was measured by quantifying its stable metabolites (NO_x_), namely the sum of nitrite (NO_2_^−^) and nitrate (NO_3_^−^), with Griess reagent as described by Miranda et al. [29]. The assay was calibrated with sodium nitrate (NaNO_3_) and NO_x_ were reported as NaNO_3_ equivalents (μM).

Advanced oxidant products (AOPP) were analyzed using citric acid following the improved method described by Hanasand et al. [30]. The assay was calibrated with chloramine-T and the results are presented as chloramine-T equivalents (μM).

Myeloperoxidase (MPO) activity was measured according to Quade and Roth [31]. The method is based on MPO-H_2_O_2_ oxidation of 3,3′,5,5′-tetramethylbenzidine hydrochloride (TMB) as a sensitive peroxidase substrate. Results are expressed as optical density measured at 450 nm (OD_450nm_).

Protease activity (PA) was determined using the azocasein hydrolysis assay, as described by Ross et al. [32], and the results are expressed as a percentage of activity in relation to the positive control, the bovine trypsin.

Anti-protease activity (APA) was determined by the capacity of the sample to inhibit trypsin activity, as described by Brunt and Austin [33]. The results are presented as percentage of inhibition of trypsin activity against the positive control, i.e., trypsin solution.

Total protein (TP) concentrations were measured by the Bradford method using a commercial Bradford reagent (cod. B6916). The assay was calibrated with bovine serum albumin (BSA) and the TP concentration is presented as BSA equivalents (mg mL^−1^).

Similar to what was reported in salivary samples collected with swabs [34], analytical results of the uterine fluid samples were normalized for the total protein content.

All the samples analyzed have been read with a microplate reader (Model 550, BioRad, Segrate, Milan, Italy) except for the TTL and AOPP assays for which a spectrophotometer (SmartSpec 3000 UV/Vis, Bio-Rad, Segrate, Italy) was used. Additional information on the analytical methods is reported in Appendix A.

### 2.9. Statistical Analysis

All data were entered into a datasheet and analyzed by Systat 11.0 (SYSTAT Software Inc., San Jose, CA, USA). Variables displaying a not normal distribution, as percentages, were transformed into angles corresponding to arcsine of the square root for variance analyses. The Shapiro–Wilks and Levene tests evaluated the normal data distribution and the homogeneity assumption needed for carrying out parametric tests. The comparison between negative and positive samples at the cytological and culture assessments either in blood serum or uterine fluid was analyzed by ANOVA. Pairwise comparison of the means was made with Fisher’s LSD test. The minimum level of statistical significance was *p* < 0.05. Values are presented as mean ± standard deviation (SD).

## 3. Results

Upon culture, 24 of the 33 mares enrolled in this study were positive for endometritis and 24 were positive for cytology. However, 6 out of 24 culture positive mares tested negative in the cytological test and, conversely, 6 cytological positive mares tested negative in the culture test. Therefore, 3 and 18 animals were negative and positive for both diagnostic tests, respectively. Only the culture positive mares were treated from day 1 to day 3 post-insemination with antibiotics according to the indications emerging from the result of the susceptibility test. As regards the type of bacteria isolated with the culture test, a combination of *Escherichia coli* (*E. coli*), *Klebsiella pneumoniae* (*K. pneumoniae*), and *Staphylococcus aureus* (*S. aureus*) was isolated in 2 mares (8.3%); a combination of *E. coli* and *K. pneumoniae* in 7 mares (29.2%); a combination of *E. coli* and *Streptococcus* group C in one mare (4.2%); only *E. coli* (33.3%) in 8 mares; only *K. pneumoniae* (8.3%) in 2 mares; only *Streptococcus* group C (8.3%) in 2 mares; and only *S. aureus* (8.3%) in 2 mares.

The two used cytological tests based on either May–Grünwald/Gimsa staining (Figure 1) or fluorescent probes (Figure 2) equally discriminated between the mares on the basis of endometritis demonstrating a full diagnostic overlap. For this reason, during the analysis of the results, these two methods have simply been indicated as the cytological test.

Table 1 reports the mean (±SD) values of the ten biomarkers analyzed in this study in either blood serum or uterine fluid samples in mares evaluated as positive and negative for endometritis on the basis of the culture test. Comparing the data of these two groups, no significant differences emerged for any of the parameters examined. Comparing, however, the same parameters in the mares discriminated as positive and negative for endometritis on the basis of the cytological test (Table 2), statistically significant differences emerged both in the blood and in the uterine fluid. In particular, the FRAP values in the blood serum were significantly higher in the endometritis-negative than -positive mares (383 ± 109 vs. 299 ± 54 µM L^−1^; *p* < 0.01) as well as, in the uterine fluid, the values of TTL (54.5 ± 17.2 vs. 34.7 ± 16.9 µM L^−1^; *p* < 0.01), NO_x_ (263 ± 126 vs. 183 ± 60 µM L^−1^; *p* < 0.05), and protease activity (13.3 ± 10.1 vs. 7.9 ± 4.9%; *p* < 0.05) were significantly higher, whereas the total protein content was significantly lower (1.98 ± 0.88 vs. 2.88 ± 1.29 mg mL^−1^, *p* < 0.05) in the endometritis-negative than in the -positive mares.

By breaking down the data distribution on the basis of simultaneous positivity and/or negativity to culture and cytological tests, four groups of mares are obtained, i.e., negative-negative (−/−, n = 3), negative-positive (−/+, n = 6), positive-negative (+/−, n = 6), and positive-positive (+/+, n = 18) to the endometritis tests. The analysis of the markers evaluated on this new distribution confirms and better outlines what has already been reported in the previous tables. In blood serum (Table 3), the lowest value of FRAP is confirmed in endometritis-positive mares. Furthermore, significant differences emerged between (−/−) and (+/−) groups in relation to the MPO values (0.144 ± 0.096 vs. 0.083 ± 0.022 OD_450nm_, *p* < 0.05). In uterine fluid (Table 4), statistically significant differences emerged between the values of TTL (58.1 ± 12.4 (−/−) and 52.6 ± 20.3 (−/+) vs. 29.2 ± 10.3 (+/−) and 36.5 ± 18.5 (+/+) µM, *p* < 0.05), NO_x_ (269 ± 83 (−/−) and 260 ± 151 (−/+) vs. 157 ± 38 (+/−) μM, *p* < 0.05) as well as of protease activity (19.2 ± 14.4 (−/−) vs. 6.4 ± 4.3 (+/−) and 8.3 ± 5.1% (+/+), *p* < 0.01), and total protein content (1.40 ± 0.42 (−/−) vs. 3.31 ± 1.75 (+/−) and 2.73 ± 1.17 (+/+) mg mL^−1^, *p* < 0.05).

Following ultrasound examination on day 15 post-insemination, a mean pregnancy rate of 36.4% (12/33) was found in the mares with a distribution of 0/3, 2/6, 2/6, and 8/18 pregnant mares in negative-negative, negative-positive, positive-negative, and positive-positive assessments based on culture and cytological tests, respectively.

Finally, the data were reprocessed by inserting positivity or negativity at the pregnancy diagnosis as a discriminating element and the results of this new processing are shown in Table 5. No statistically significant differences emerged between the various parameters examined when not evaluated on either blood serum or on uterine fluid.

## 4. Discussion

The analysis of some parameters related to redox balance, inflammation, and protease regulator potential in blood serum and uterine fluid allowed for discrimination between mares diagnosed as endometritis-positive and -negative by cytological test. Conversely, the culture test did not highlight a significative discrimination between these groups. These results support the finding of numerous authors about the lower reliability of the culture compared with the cytological test [11,12,35] as well as the sensitivity of the parameters evaluated in the disclosure of endometritis. In spite of these diagnostic results, however, it should be noted that the pregnancy rate did not show significant variations either in relation to the different diagnostic tests used or when referred only to the positive and negative mares for both diagnostic tests. Moreover, the high incidence of endometritis recorded in this study, when referring to both the cytological test (72.7%) and the culture test (72.7%), is remarkable when compared to other authors who report an incidence of bacterial endometritis in mares ranging between 25 and 60% (for review see [8]). However, other researchers report higher values reaching 66.2% [36] and, recently, up to 92.6% [37]. Further, when these two diagnostic tests were combined, the incidence becomes even alarming (30/33 = 90.9%). We do not have a clear explanation for this finding, which can be associated with management factors, such as poor hygiene [38], as well as an increasing incidence of this pathology within the population of mares [37] or an increase in diagnostic capacity resulting from the use of enriched culture media [21].

The cytological test by fluorescence microscopy was an upgrade of a methodology previously proposed using the fluorescence spectroscopy [14] and slightly modified in the present study. It proved to be as equally reliable as the classic cytological test with the advantage of simplicity of preparation and reading.

This study aimed to explore alternative endometritis diagnostics in the mare based on blood tests. This possibility is specially required in mares because pre-existing conditions of endometrial infections/inflammation can be further exacerbated by the PBE. Treatments for solving this latter occurrence can be performed immediately after insemination. However, their follow-up represents a diagnostic problem due to the impossibility of direct access to the uterus to avoid repercussions on the development of pregnancy. Indirect information on the uterus would be interesting and useful for monitoring the post-treatment response of the uterus in inseminated mares. In this study, we analyzed, in blood serum and uterine fluid samples, biomarkers either related to OS, such as TOS, NOx, and AOPP, or antioxidant activity, such as FRAP, TAC, FRSA, TTL as well as to inflammation, such as MPO, and protease regulator potential, such as PA and APA. Of these ten parameters examined in blood serum, however, only FRAP shows significant differences between cytologically endometritis-positive and -negative mares. FRAP evaluates the combined activity of a set of substances with antioxidant activity that an organism has to counteract during an increase in oxidizing substances, deriving from numerous occurrences including pathologies. Whole antioxidant evaluations represent assessments of the overall antioxidant status, summing up interactions between the different antioxidant molecules. Usually, the antioxidant potential of a biological sample is measured as the content of free radicals scavenged by a test compound [39]. Hence, the value resultant from each test represents the integrated cumulative action of those antioxidant substances of the biological sample reacting with the test compound. This means that the results obtained are often not comparable between tests. For example, the ferric reducing ability of plasma (FRAP) assay [23] is sensitive to uric acid, representing approximately 60% of the antioxidant capacity in human samples [40], as well as to low molecular weight antioxidants, such as αtocopherol, bilirubin, and ascorbic acid; however, it is not sensitive to antioxidants with plenty of SH groups, such as glutathione (GSH) and albumin. The total antioxidant capacity (TAC) assay, based on the ABTS method, similar to the FRAP assay, is affected by the uric acid, even if less than in the FRAP assay, and by low molecular weight antioxidants; however, differently to the FRAP assay, it is also affected by albumin, GSH, and phenolic compounds [41]. Total thiol level (TTL) assay is the method of choice to assess the antioxidant barrier ensured by thiol protein groups, as found in donkey semen [41]. Hence, each test used for evaluating the oxidant/antioxidant content of a biological sample involves numerous substances endowed with these properties to a different extent and this inevitably leads to incomparable, and sometimes, unrelated results. For this reason, a combined use of different assays is required to get reliable information on the redox balance. However, the studies conducted on the redox balance are often incomplete, analyzing only a partial side of the mechanisms involved and leaving out an overall view. In line with this, queens affected by pyometra showed higher serum levels of acute phase proteins and FRAP values, together with lower values of total serum thiols; this last finding is an expression of an SH-group-mediated antioxidant barrier and is associated with lower albumin serum content [42]. In this study, however, the total oxidant status of blood serum was not assessed; nevertheless, based on the increase of acute phase protein serum concentration and the antioxidant imbalance, the authors concluded that queens with pyometra experienced OS.

An increase in oxidant levels may be counteracted by an increase in the antioxidant barrier, maintaining an unchanged oxidant/antioxidant balance, as during the peripartum in the mares [43]; this resilience to OS is named eustress (OeS). Conversely, when OS cannot be efficiently contrasted and the oxidative damage of macromolecules becomes an irreversible occurrence, the organism undergoes a condition of distress (OdS) [18]. In the present study, the enhancement of the antioxidant barrier, as indicated by the increase in FRAP and TTL values, may be associated with the increase in NO_x_, which was significantly higher in the endometritis-negative than in the -positive mares, and aimed at maintaining the correct redox balance. Reactive nitrogen species (RNS), which are detected as NO_x_, are involved in many physiological activities [44]. Recently, NO_x_ has been found positively related to FRAP levels and MPO activity as well as to both natural antibody IgM-isotype (Nab-IgM) and total immunoglobulin IgM-isotype (tot Ig-M) levels in heathy goat kids [45,46]. Besides their noxious effects on the body homeostasis, in mice, high NO levels seem to be involved in the infection control before occurring in adaptive immunity [47], with a pivotal role in regulating the activity of the different B-cell lines [46]. However, in mares affected by endometritis, recent studies by Abdelnaby et al. [15] showed a significant increase in malondialdehyde (MDA) and NO metabolites levels associated with a significant decrease in TAC in comparison with healthy mares; these findings suggest that this group of mares experienced OS. Instead, the lack of significant variations in AOPP levels, marking the occurrence of oxidative protein damage, suggests that our group of mares affected by endometritis have not experienced OS. In addition, some concerns can be raised about the research of Abdelnaby and colleagues [15], which do not describe in detail the methodology used to analyze TAC and NO metabolites. For the latter, in particular, it is not reported whether the method included only the amount of nitrite or, as for the method used by us, the sum of nitrite and nitrate contained in the samples. This does not allow comparison between the results obtained in this research with those of the present study.

The MPO, an inflammation marker and a prooxidant enzyme contained in and released by neutrophils during degranulation or after cell lysis [48], did not show significant variations between the endometritis-negative and -positive mares except for the (−/−) and (+/−) groups at cytological and culture tests, respectively. In mares, a previous study [49] found a significant increase in MPO levels in uterine fluid samples in cytologically endometritis-positive subjects analyzed by ELISA. In contrast, it found no differences when endometritis was based on clinical findings, such as uterine hyperedema and the presence of intrauterine fluid. Although the significant differences between cytologically endometritis-positive and -negative subjects, the extreme variability detected by this analytical method and the high incidence of false positives made this examination poorly reliable.

The total protein content of uterine fluid samples was significantly higher in cytologically endometritis-positive mares. This result represents another element in support of the greater reliability of the cytological compared to the culture test considering the predictable result as already detected in other biological fluids, such as saliva, in conditions of inflammation/infections [50].

Regarding the protease regulator potential, the present study showed significantly higher values of protease activity in the uterine fluid of healthy than in cytologically endometritis-positive mares. Protease regulators is a complex of proteins and enzymes with protease and protease inhibitor activities that are involved in several biological aspects, including infectious defense and reproductive biology [41,51]. For an overall assessment of these activities in blood and uterine fluid, we applied two spectrophotometric methods that have been previously used in semen plasma to predict sperm functionality in donkeys [41]. In this study, we demonstrated that these simple methods can be also applied to the blood serum and uterine fluid of mares for detecting reliable findings on the defense ability against pathogens.

## 5. Conclusions

In mares, a panel of ten markers of redox balance, inflammation, and protease regulator potential was used on blood serum and uterine fluid samples and related to endometritis that had been evaluated before insemination with both cytological and culture tests. In blood, only one of these markers, the FRAP, which estimates the antioxidant power in organic fluids, was able to significantly discriminate between cytologically endometritis-positive and -negative mares. In the uterine fluid, other markers significantly discriminated between these animals, such as TTL, a marker of the antioxidant power, NO_x_, a marker of the nitric oxide metabolites, as well as the total protein content and the protease activity, a parameter involved in multiple biological and inflammatory processes. The analysis of these variables should be considered as a whole in the variegated expression of the redox potential, showing an equilibrium or an imbalance. Although not providing decisive diagnostic indications, the evaluation of these parameters allows us to obtain useful information for the study of the endometritis dynamics. This exploration study, therefore, constitutes a necessary premise to focus attention on the identification of indirect markers of endometritis which, through a blood test, can provide reliable diagnostic indications on this pathology.

## Figures and Tables

**Figure 1 animals-13-00253-f001:**
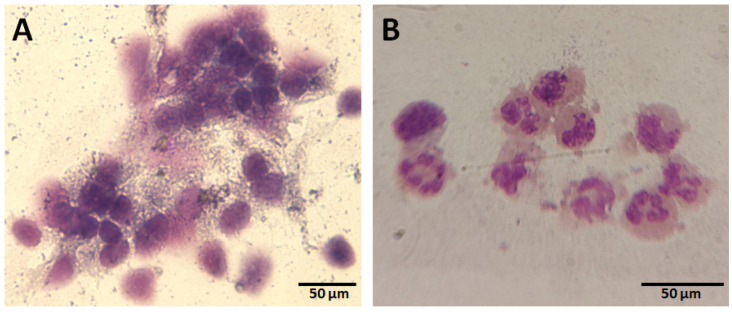
Representative images of cytological assessment by May–Grünwald/Gimsa staining of uterine cell smears in endometritis negative (**A**) and positive (**B**) mares.

**Figure 2 animals-13-00253-f002:**
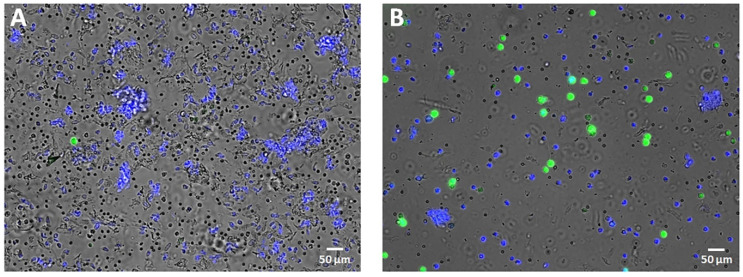
Representative images of cytological assessment of uterine cells stained with H_2_DCF-DA and H33342 and analyzed by fluorescence microscopy in endometritis negative (**A**) and positive (**B**) mares.

**Table 1 animals-13-00253-t001:** Mean (±SD) values of oxidative, inflammation, and protease regulator biomarkers assessed in the blood serum and the uterine fluid of mares with negative (n = 9) and positive (n = 24) outcomes for the diagnosis of endometritis based on microbiological assessment.

Microbiological Assessment Biomarkers	Blood Serum		Uterine Fluid	
	Negative Mean ± SD	Positive Mean ± SD	Negative Mean ± SD	Positive Mean ± SD
FRAP (FeSO_4_•7H_2_O equivalents, μM)	338 ± 111	316 ± 69	53.5 ± 14.5	54.0 ± 21.0
TAC (AA equivalents, μM)	0.651 ± 0.268	0.602 ± 0.361	0.016 ± 0.034	0.031 ± 0.091
FRSA (AA equivalents, mg mL^−1^)	2.37 ± 0.44	2.51 ± 0.59	0.940 ± 0.339	1.081 ± 0.502
TTL (µM)	205 ± 25	221 ± 23	38.8 ± 17.7	40.5 ± 19.8
TOS (*t*-BHP equivalents, μM)	8.40 ± 9.50	13.48 ± 13.79	3.94 ± 5.48	3.85 ± 3.99
NO_x_ (NaNO_3_ equivalents, μM)	197 ± 103	201 ± 113	194 ± 76	208 ± 95
AOPP (chloramine-T equivalents, μM)	11.0 ± 2.3	13.4 ± 8.6	13.9 ± 4.7	11.9 ± 6.8
MPO (optical density, OD_450nm_)	0.104 ± 0.060	0.102 ± 0.034	0.015 ± 0.009	0.015 ± 0.009
PA (trypsin activity, %)	14.8 ± 6.3	12.5 ± 7.6	10.6 ± 10.2	8.86 ± 5.53
APA (trypsin activity inhibition,%)	95.2 ± 2.0	95.8 ± 1.9	7.48 ± 4.50	5.98 ± 5.46
Total proteins (mg mL^−1^)	64.4 ± 1.5	65.0 ± 1.7	2.66 ± 1.71	2.63 ± 1.06

FRAP: total antioxidant capacity; TAC: total antioxidant capacity; FRSA: free radical scavenging activity; TTL: total thiol levels; TOS: total oxidant status; NOx: nitric oxide metabolites; AOPP: advanced oxidation protein products; MPO: Myeloperoxidase; PA: protease activity; APA: antiprotease activity; AA: ascorbic acid.

**Table 2 animals-13-00253-t002:** Mean (±SD) values of oxidative, inflammation, and protease regulator biomarkers assessed in the blood serum and the uterine fluid of mares with negative (n = 9) and positive (n = 24) outcomes for the diagnosis of endometritis based on cytological assessment.

Cytological Assessment Biomarkers	Blood Serum		Uterine Fluid	
	Negative Mean ± SD	Positive Mean ± SD	Negative Mean ± SD	Positive Mean ± SD
FRAP (FeSO_4_•7H_2_O equivalents, μM)	383 ± 109 ^A^	299 ± 54 ^B^	52.8 ± 16.9	54.2 ± 20.4
TAC (equivalents, μM)	0.748 ± 0.377	0.565 ± 0.311	0.018 ± 0.040	0.030 ± 0.090
FRSA (AA equivalents, mg mL^−1^)	2.51 ± 0.62	2.46 ± 0.54	1.233 ± 0.437	0.971 ± 0.460
TTL (µM)	210 ± 27	219 ± 23	54.5 ± 17.2 ^A^	34.7 ± 16.9 ^B^
TOS (*t*-BHP equivalents, μM)	9.60 ± 11.30	13.02 ± 13.47	3.416 ± 4.055	4.048 ± 4.536
NO_x_ (NaNO_3_ equivalents, μM)	186 ± 57	205 ± 124	263 ± 126 ^a^	183 ± 60 ^b^
AOPP (chloramine-T equivalents, μM)	14.3 ± 12.6	12.2 ± 4.5	13.5 ± 5.4	12.0 ± 6.7
MPO (optical density, OD_450nm_)	0.127 ± 0.054	0.094 ± 0.033	0.014 ± 0.008	0.015 ± 0.010
PA (trypsin activity, %)	13.9 ± 7.1	12.9 ± 7.4	13.3 ± 10.1 ^a^	7.85 ± 4.88 ^b^
APA (trypsin activity inhibition,%)	95.8 ± 1.2	95.6 ± 2.1	6.64 ± 4.51	6.35 ± 5.44
Total proteins (mg mL^−1^)	64.3 ± 1.5	65.0 ± 1.6	1.98 ± 0.88 ^a^	2.88 ± 1.29 ^b^

For the complete legend see Table 1. ^A,B^
*p* < 0.01; ^a,b^
*p* < 0.05.

**Table 3 animals-13-00253-t003:** Mean (±SD) values of oxidative, inflammation, and protease regulator biomarkers assessed in the blood serum of mares with negative and positive outcomes for the diagnosis of endometritis based on either microbiological or cytological assessments.

Cytological Test Culture Test	−	−	+	+
	−	+	−	+
n. Biomarkers	3 Mean ± SD	6 Mean ± SD	6 Mean ± SD	18 Mean ± SD
FRAP (FeSO_4_•7H_2_O equivalents, μM)	409 ± 182 ^a^	370 ± 73 ^a^	302 ± 43 ^b^	298 ± 59 ^b^
TAC (AA equivalents, μM)	0.661 ± 0.332	0.792 ± 0.420	0.645 ± 0.266	0.539 ± 0.327
FRSA (AA equivalents, mg mL^−1^)	2.55 ± 0.25	2.49 ± 0.76	2.28 ± 0.51	2.52 ± 0.55
TTL (µM)	199 ± 19	215 ± 30	208 ± 28	223 ± 21
TOS (*t*-BHP equivalents, μM)	13.9 ± 15.1	7.43 ± 9.79	5.62 ± 5.05	15.5 ± 14.6
NO_x_ (NaNO_3_ equivalents, μM)	204 ± 57	177 ± 60	193 ± 125	209 ± 127
AOPP (chloramine-T equivalents, μM)	11.9 ± 2.8	15.5 ± 15.7	10.5 ± 2.2	12.7 ± 4.9
MPO (optical density, OD_450nm_)	0.144 ± 0.096 ^a^	0.118 ± 0.025	0.083 ± 0.022 ^b^	0.097 ± 0.036
PA (trypsin activity, %)	13.5 ± 7.3	14.0 ± 7.8	15.4 ± 6.4	12.0 ± 7.7
APA (trypsin activity inhibition,%)	96.4 ± 0.4	95.5 ± 1.4	94.5 ± 2.2	95.9 ± 2.0
Total proteins (mg mL^−1^)	64.7 ± 1.9	64.1 ± 1.5	64.3 ± 1.4	65.3 ± 1.7

For the complete legend see Table 1. ^a,b^ (*p* < 0.05).

**Table 4 animals-13-00253-t004:** Mean (±SD) values of oxidative, inflammation, and protease regulator biomarkers assessed in the uterine fluid of mares with negative and positive outcomes for the diagnosis of endometritis based on either microbiological or cytological assessments.

Cytological Test Culture Test	−	−	+	+
	−	+	−	+
n. Biomarkers	3 Mean ± SD	6 Mean ± SD	6 Mean ± SD	18 Mean ± SD
FRAP (FeSO_4_•7H_2_O equivalents, μM)	59.4 ± 19.8	49.6 ± 16.2	50.5 ± 12.3	55.5 ± 22.6
TAC (AA equivalents, μM)	0.014 ± 0.024	0.020 ± 0.049	0.016 ± 0.040	0.035 ± 0.102
FRSA (AA equivalents, mg mL^−1^)	1.25 ± 0.30	1.22 ± 0.52	0.78 ± 0.24	1.03 ± 0.50
TTL (µM)	58.1 ± 12.4 ^a^	52.6 ± 20.3 ^a^	29.2 ± 10.3 ^b^	36.5 ± 18.5 ^b^
TOS (*t*-BHP equivalents, μM)	2.82 ± 4.88	3.72 ± 4.06	4.50 ± 6.12	3.90 ± 4.09
NO_x_ (NaNO_3_ equivalents, μM)	269 ± 83 ^a^	260 ± 151 ^a^	157 ± 38 ^b^	191 ± 65
AOPP (chloramine-T equivalents, μM)	17.3 ± 4.8	11.5 ± 5.0	12.1 ± 3.9	12.0 ± 7.5
MPO (optical density, OD_450nm_)	0.015 ± 0.011	0.014 ± 0.008	0.015 ± 0.010	0.015 ± 0.010
PA (trypsin activity, %)	19.2 ± 14.4 ^A,a^	10.4 ± 7.0 ^b^	6.4 ± 4.3 ^B^	8.3 ± 5.1 ^B^
APA (trypsin activity inhibition,%)	10.3 ± 4.6	3.9 ± 1.8	7.3 ± 4.1	7.2 ± 5.9
Total proteins (mg mL^−1^)	1.40 ± 0.42 ^a^	2.30 ± 0.90	3.31 ± 1.75 ^b^	2.73 ± 1.17 ^b^

For the complete legend see Table 1. ^A,B^
*p* < 0.01; ^a,b^
*p* < 0.05.

**Table 5 animals-13-00253-t005:** Mean (±SD) values of the oxidative profile and proteolysis regulator biomarkers assessed in the blood serum and the uterine fluid of mares with negative (n = 21) and positive (n = 12) outcomes for the pregnancy diagnosis.

Outcome of Pregnancy Diagnosis Biomarkers	Blood Serum		Uterine Fluid	
	Negative Mean ± SD	Positive Mean ± SD	Negative Mean ± SD	Positive Mean ± SD
FRAP (FeSO_4_•7H_2_O equivalents, μM)	337 ± 92	296 ± 51	56.9 ± 21.7	48.5 ± 13.2
TAC (AAequivalents, μM)	0.59 ± 0.34	0.66 ± 0.34	0.033 ± 0.097	0.016 ± 0.033
FRSA (AA equivalents, mg mL^−1^)	2.53 ± 0.56	2.37 ± 0.55	1.057 ± 0.497	1.018 ± 0.415
TTL (µM)	216 ± 26	216 ± 21	39.3 ± 17.6	41.3 ± 22.0
TOS (*t*-BHP equivalents, μM)	11.0 ± 8.4	14.0 ± 18.5	2.89 ± 3.56	5.60 ± 5.20
NO_x_ (NaNO_3_ equivalents, μM)	178 ± 85	236 ± 138	195 ± 70	220 ± 117
AOPP (chloramine-T equivalents, μM)	13.6 ± 9.0	11.3 ± 3.2	11.7 ± 6.5	13.6 ± 6.0
MPO (optical density, OD_450nm_)	0.109 ± 0.049	0.092 ± 0.020	0.014 ± 0.008	0.016 ± 0.011
PA (trypsin activity, %)	13.7 ± 7.5	12.1 ± 6.9	9.32 ± 6.99	9.38 ± 7.28
APA (trypsin activity inhibition, %)	95.5 ± 1.7	95.9 ± 2.2	6.18 ± 6.11	6.85 ± 3.01
Total proteins (mg mL^−1^)	64.6 ± 1.6	65.2 ± 1.7	2.45 ± 1.27	2.95 ± 1.17

For the complete legend see Table 1.

## Data Availability

Not applicable.

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
