# Peer review of "Serological and Uterine Biomarkers for Detecting Endometritis in Mares"

_animals, 2023, doi:10.3390/ani13020253_

Round 1
Reviewer 1 Report
This study tried to to explore alternative endometritis diagnostics in the mare based 342 on blood tests. But the chosen PMN concentration threshold is too low and there is probably a heavy contamination of the samples.
Timing of bacteriological and antibiotic susceptibility results should be explained more accurately with a lab certification.
Seems that this study has been performed by researchers not really into equine reproduction even though the starting idea was really interesting.
I suggest to perform this protocol in a controlled context under supervision of an expert.
110. You should describe how you detected estrus, what was the follicular diameter at ovulation induction with hCG.
115. Explain how can you have had a susceptibility test result in 48 hours in a commercial context? These sentences seem to contradict with lines 147-153. Provide the laboratory name and certifications. Moreover, a single flushing (1 liter) isn't the gold standard to remove the inflammatory elements, provide literature in support.
161 “The finding of one or more PMNs on a hundred counted cells labeled the cell sample as belonging to an endometritis-positive mare.”
Provide literature supporting a so low number of PMN's are enough to diagnose endometritis. Usually a 2-5% cutoff is considered the lower limit.
238: Any suspected bacterial contamination of the sample? The numbers are not consistent with literature findings.
Author Response
This study tried to to explore alternative endometritis diagnostics in the mare based 342 on blood tests. But the chosen PMN concentration threshold is too low and there is probably a heavy contamination of the samples.
Reply. We thank the reviewer for the careful analysis conducted which will certainly allow an improvement in the quality of our paper. With regards to the PMN threshold level, it is a puzzle, considering that almost every author has his own standard referring to the number of PMNs per microscopic field at different magnifications or on the total counted cells (for review see Card, Theriogenology 2005 - 10.1016/j.theriogenology.2005.05.041). Based on the number of PMN on the total number of counted uterine cells, the PMN threshold used in the present study complies with that of many other studies conducted in the equine species that set this value between 0.5% (10.1016/j.jevs.2009.11.006; 10.1016/j.anireprosci.2012.05.012; 10.1016/j.prevetmed.2022.105783) and 2% (10.1016/0093-691X(88)90007-6; 10.1016/j.theriogenology.2010.12.002). A 1% PMN rate seemed to us a good compromise as suggested by several authors (10.1016/S0737-0806(89)80020-6; 10.1016/j.theriogenology.2005.05.041; 10.1016/j.theriogenology.2007.04.038). Certainly, these rates are lower than those set in the bovine species in which they vary from 5 to 8%. In any case, the use of the double-guarded uterine swabs has been conducted with rigorous accuracy and we exclude any contamination of the collected material.
Timing of bacteriological and antibiotic susceptibility results should be explained more accurately with a lab certification.
Reply. We apologize for not having elaborated on this aspect. We have, therefore, added further information on the susceptibility (or antibiogram) test method (L164-169). All analyzes (microbiological, cytological, and biochemical) were carried out in the laboratories of the Department of Sciences, University of Basilicata, and we added this information to the text (L146-147). These laboratories are used only for research activities; hence, accreditation certifications are not required as for private public service laboratories. Of course, our laboratories are certified by an internal assessment service with regard to chemical, physical and biological risk assessments.
Seems that this study has been performed by researchers not really into equine reproduction even though the starting idea was really interesting. I suggest to perform this protocol in a controlled context under supervision of an expert.
Reply. We thank the reviewer for appreciating our study. Two of the authors (S.C.G. and R.F.) are DVM equine neophytes, with little work in donkeys and mares but more than 30 years of lab work in reproductive biology (R.B.) and immunology and oxidative stress evaluation (S.C.G.) The third author (T.D.P.) is a DVM freelancer who works for more than one decade in equine reproduction with few scientific papers but very high professional skills and extreme attention to updates in the field of equine reproduction.
- You should describe how you detected estrus, what was the follicular diameter at ovulation induction with hCG.
Reply. We reported this information in the text (L113-114).
- Explain how can you have had a susceptibility test result in 48 hours in a commercial context? These sentences seem to contradict with lines 147-153. Provide the laboratory name and certifications. Moreover, a single flushing (1 liter) isn't the gold standard to remove the inflammatory elements, provide literature in support.
Reply. As we have already explained above in relation to the laboratories, the analyzes performed did not fall within a commercial context but within a scientific study. However, you are right, sorry, 48 h is the shortest time required in the case of Gram- infections (12 h culture in enrichment broth + 24 h in MacConkey + 12 h for the susceptibility test). In the case of Gram+ infections, the time can increase up to 96 h. This means that the mares with uterine infections can be treated with a tested antibiotic from 1 to 3 days post-insemination. We added this range in the text (L118). Our co-author (T.D.P.) ensures these times are assured also in the normal operational routine. For susceptibility test and labs, see above. Regarding washing with 1 L of saline or Ringer’s lactate solution, we added the reference (Maischberger, et al., 2008).
161 “The finding of one or more PMNs on a hundred counted cells labeled the cell sample as belonging to an endometritis-positive mare.”
Provide literature supporting a so low number of PMN's are enough to diagnose endometritis. Usually a 2-5% cutoff is considered the lower limit.
Reply. We have already responded to this criticism above.
238: Any suspected bacterial contamination of the sample? The numbers are not consistent with literature findings.
Reply. As we reported above, all the sampling procedures were carried out with absolute scientific rigor. Regarding the high number of positives, it seemed high to us too; however, this is the result obtained and it is supported by other studies and commented on in Discussion (L349-358).
Reviewer 2 Report
Dear editors,
this paper investigates if antioxidant parameter concentration in the blood serum can be taken into account as diagnostic markers to identify mares affected by endometritis. The authors compared the antioxidant status in clinically healthy Quarter horse mares divided based on positivity or negativity for endometritis, detected with cytologic and culture methods. They highlighted that the ferric reducing ability of plasma (FRAP) assay, a marker of the antioxidant power, allowed to significantly discriminate the mares diagnosed cytologically positive and negative for endometritis. The topic is very interesting and current. The manuscript is well written and structured. The references are specific and relevant.
Summary and Abstract: These sections are well written and recap the information contained in the main text without repetitions. However, the abstract should contain the question addressed in a broad context highlighting the purpose of the study. I suggest to introduce this part, reminding that the maximum number of words is about 200 words for this section. In addition, beside indicating that there is a difference between the groups, you should specify that FRAP resulted significantly higher in the endometritis-negative than -positive mares.
Key-words: they are pertinent and consistent with the topic. You could add “ferric reducing ability of plasma (FRAP)”, since it is the parameter which resulted significantly higher in the endometritis-negative than -positive mares on the basis of the cytological test.
Introduction: This section properly shows the state-of-the-art resuming the knowledge about this topic. The aim of the study is discussed but the hypothesis is not clearly expressed.
Methods and Results: These sections are well structured and performed. The section “Methods” properly describes the sample sizes, the procedures and statistical tests. However, the section 2.4 “Collection of blood and uterine fluid samples” does not contain the blood sampling protocol. Please, add it. I think that the section 2.1 “Reagents“ could be suppressed, adding these individual information where the reagents appear along the text for the first time. I have some questions: Did you recollect the blood samples at the same hour for all subjects? Maybe the time was different if the sampling has been done at the moment of the heat signs. Were the mares fed with the same diet? Diet components with different concentration of antioxidants can affect the redox balance. All these aspects should be discussed. Results are well presented and joined with very detailed tables and attractive figures, adding useful information to the main text.
Discussion: this section is logically written presenting your persuasive interpretation. However, the limits of this study should be clearly presented. Ref. 39: you should specify that the paper is referring to donkeys. Ref. 45-46: it would be better to add the species these papers are referring to. In general, this should be done along the text, wherever this detail is missing.
Author Response
Dear editors,
this paper investigates if antioxidant parameter concentration in the blood serum can be taken into account as diagnostic markers to identify mares affected by endometritis. The authors compared the antioxidant status in clinically healthy Quarter horse mares divided based on positivity or negativity for endometritis, detected with cytologic and culture methods. They highlighted that the ferric reducing ability of plasma (FRAP) assay, a marker of the antioxidant power, allowed to significantly discriminate the mares diagnosed cytologically positive and negative for endometritis. The topic is very interesting and current. The manuscript is well written and structured. The references are specific and relevant.
Reply. We enjoy that the reviewer appreciated our study and we thank her/him for her/his words.
Summary and Abstract: These sections are well written and recap the information contained in the main text without repetitions. However, the abstract should contain the question addressed in a broad context highlighting the purpose of the study. I suggest to introduce this part, reminding that the maximum number of words is about 200 words for this section. In addition, beside indicating that there is a difference between the groups, you should specify that FRAP resulted significantly higher in the endometritis-negative than -positive mares.
Reply. Done (L23 and L30-32).
Key-words: they are pertinent and consistent with the topic. You could add “ferric reducing ability of plasma (FRAP)”, since it is the parameter which resulted significantly higher in the endometritis-negative than -positive mares on the basis of the cytological test.
Reply. Done.
Introduction: This section properly shows the state-of-the-art resuming the knowledge about this topic. The aim of the study is discussed but the hypothesis is not clearly expressed.
Reply. Done (L91-95).
Methods and Results: These sections are well structured and performed. The section “Methods” properly describes the sample sizes, the procedures and statistical tests. However, the section 2.4 “Collection of blood and uterine fluid samples” does not contain the blood sampling protocol. Please, add it.
Reply. Sorry for this missing information, we added it to the text (L133-135 and L147-149).
I think that the section 2.1 “Reagents“ could be suppressed, adding these individual information where the reagents appear along the text for the first time.
Reply. We followed the Reviewer’s suggestion but maintained this Section to avoid repeating Sigma provider for the many products referring to this company.
I have some questions:
Did you recollect the blood samples at the same hour for all subjects? Maybe the time was different if the sampling has been done at the moment of the heat signs.
Reply. The animals were visited in this breeding station in the morning. So, all the operations were conducted between 8:00 and 12:00.
Were the mares fed with the same diet? Diet components with different concentration of antioxidants can affect the redox balance. All these aspects should be discussed.
Reply. Yes, all animals were fed the same way (L108-111), and, yes, we are aware that diet can influence the redox balance, as highlighted in some previously conducted studies (10.3168/jds.2018-15857; 10.17221/189/2019-CJAS; 10.3168/jds.2014-8414). Therefore, we were careful that this source of variation was under control.
Results are well presented and joined with very detailed tables and attractive figures, adding useful information to the main text.
Reply. We thank the reviewer for the appreciation of our study and for the numerous insights provided.
Discussion: this section is logically written presenting your persuasive interpretation. However, the limits of this study should be clearly presented. Ref. 39: you should specify that the paper is referring to donkeys. Ref. 45-46: it would be better to add the species these papers are referring to. In general, this should be done along the text, wherever this detail is missing.
Reply. Done.
Reviewer 3 Report
This manuscript was identified some parameters in blood and uterine fluid of mares associated with redox balance, inflammation, and protease regulator potential and possible markers of endometritis using bacteriological, cytological and biochemical analyses. The authors found that total antioxidant capacity (FRAP) in the blood serum, total thiol levels (TTL), nitric oxide metabolites (NOx), protease activity and total protein content in the uterine fluid, allowed to significantly discriminate between cytologically endometritis-positive and -negative mares. The results of this manuscript are scientifically and clinically important for diagnosis of endometritis in the mare.
This manuscript is well organized structure and appropriated figures. However, there are still some suggestions.
Sample sizes/group sizes (or number of repeats) should be given in the Materials and Methods section.
have not described how to collection of blood samples.
Table1: The font in the first row should be uniform.
Why treated with antibiotics? There is no significant correlation between the biomarkers and pregnancy rate, whether it is caused by antibiotic treatment?
Author Response
This manuscript was identified some parameters in blood and uterine fluid of mares associated with redox balance, inflammation, and protease regulator potential and possible markers of endometritis using bacteriological, cytological and biochemical analyses. The authors found that total antioxidant capacity (FRAP) in the blood serum, total thiol levels (TTL), nitric oxide metabolites (NOx), protease activity and total protein content in the uterine fluid, allowed to significantly discriminate between cytologically endometritis-positive and -negative mares. The results of this manuscript are scientifically and clinically important for diagnosis of endometritis in the mare.
This manuscript is well organized structure and appropriated figures. However, there are still some suggestions.
Reply. We thank the reviewer for the appreciation of our study and for the numerous insights provided.
Sample sizes/group sizes (or number of repeats) should be given in the Materials and Methods section.
have not described how to collection of blood samples.
Reply. Regarding blood samples, sorry for this missing information, we added it to the text (L133-135 and L147-149). Regarding sample sizes/group sizes (or number of repeats), we referred to the total number of mares enrolled in this study (L104). The subsequent groupings are derived from the results of the culture and cytological analyzes or based on the insemination outcome (pregnant/empty); this did not allow the creation of groups in the experimental design phase or referable to the M&M section.
Table1: The font in the first row should be uniform.
Reply. Done
Why treated with antibiotics? There is no significant correlation between the biomarkers and pregnancy rate, whether it is caused by antibiotic treatment?
Reply. Treatment with antibiotics was carried out only in the presence of a positive finding of the culture test and aimed at resolving the uterine infection. This is usually the first therapeutic aid, then, if the infection does not resolve, the clinician switches to other therapeutic aids ranging from topical disinfectants to ozone therapy. For this study, however, we only referred to the first insemination and antibiotic treatment alone. Regarding the possible interference of antibiotic treatment on pregnancy and, hence, the side effects of antibiotics, this is a topic that divides medicine not only into reproduction but extended into all pathological forms.
Reviewer 4 Report
Equipment used for the biochemical analyses needs to be described not just the technique.
Information of the precision of the methods for the samples studies and for each parameter are needed, as part of the validation procedure of such determinations in uterine fluids and blood.
Figure 1 A and B, quality of the figures are low, if it is possible they should be improved.
Discussion is well organized and cover relevant findings of the study.
Author Response
Equipment used for the biochemical analyses needs to be described not just the technique.
Reply. We added this information (L244-247)
Information of the precision of the methods for the samples studies and for each parameter are needed, as part of the validation procedure of such determinations in uterine fluids and blood.
Reply. We added additional information on analytic methods in Supplement 1.
Figure 1 A and B, quality of the figures are low, if it is possible they should be improved.
Reply. Sorry, we cannot resolve the reviewer’s criticism. The quality of the images is the best we can make based on the microscope and camera available and cytological preparation based on uterine swab-collected cell smears. However, the quality of our images complies with that of other studies on uterine cytology in the literature (https://doi.org/10.3390/ani10061062; https://doi.org/10.1016/j.theriogenology.2011.07.020; https://doi.org/10.1111/eve.12280)
Discussion is well organized and cover relevant findings of the study.
Reply. We thank the reviewer for the appreciation of our study.